# In Vitro Activity of Cefiderocol Against Multi-Drug-Resistant Gram-Negative Clinical Isolates in Romania

**DOI:** 10.3390/antibiotics14111113

**Published:** 2025-11-05

**Authors:** Anda Baicus, Valentina Daniela Sisu, Andreia Domnica Tatar, Adrian Gherghel, Diana Gabriela Iacob, Corina Silvia Pop, Catalin Florin Cirstoiu

**Affiliations:** 1Emergency University Hospital, 050098 Bucharest, Romania; 2University of Medicine and Pharmacy, Carol Davila, 020021 Bucharest, Romania; 3National Institute of Research and Development for Microbiology and Immunology Cantacuzino, 050096 Bucharest, Romania

**Keywords:** multidrug-resistant (MDR) bacteria, cefiderocol, carbapenemases

## Abstract

**Background/Objectives**: Gram-negative bacteria with acquired carbapenem resistance have become increasingly common in serious infections. This study aimed to evaluate the in vitro activity of cefiderocol against multidrug-resistant Gram-negative clinical isolates collected from a tertiary care hospital in Romania. **Methods**: A retrospective study (November 2024–February 2025) involving 89 consecutive Multi-Drug Resistant (MDR) Gram-negative isolates, 66 *Enterobacterales* and 23 non-fermenters (*P. aeruginosa*, *S. maltophilia*, *A. baumannii*), was conducted. **Results**: Overall, 52.8% of *Enterobacterales* and *P. aeruginosa* isolates were susceptible to cefiderocol, with higher susceptibility among *Enterobacterales* alone (63.6%). Using disk diffusion, 66.3% of all isolates were classified as susceptible to the antibiotic. *K. pneumoniae* isolates co-harboring NDM and OXA-48 were susceptible in 65.3% of cases, while NDM-only producers were resistant. All *P. aeruginosa* isolates tested were susceptible to the antibiotic. Susceptibility rates in *A. baumannii* were lower (68.8%), with variability between testing methods. **Conclusions**: The presence of NDM-producing isolates with complete resistance to cefiderocol in our study highlights the risk that resistance may spread rapidly once the drug becomes widely used. Cefiderocol may be an effective option for treating MDR bacterial infections, but strict microbiological monitoring remains essential.

## 1. Introduction

The increasing spread of Multi-Drug Resistant (MDR) bacteria is a major concern for society because it results in higher rates of illness and death.

Risk factors for colonization or infection with MDR bacteria include prolonged treatment with broad-spectrum cephalosporins and/or carbapenems, prior antibiotic use, severe illness, diabetes, malignancy, mechanical ventilation, and indwelling urinary or venous catheters [1].

In healthcare settings, Carbapenemase-Producing Organisms (CPOs), such as *Enterobacterales* strains and opportunistic bacteria like *Pseudomonas aeruginosa* and *Acinetobacter baumannii*, are considered superbugs because they have lost susceptibility to almost all traditional antibiotic classes, including carbapenems.

In 2017, the WHO designated carbapenem-resistant (CR) *Enterobacteriaceae*, CR *Pseudomonas aeruginosa*, and CR *Acinetobacter baumannii* as the top priority group for developing new and effective antibiotic treatments [2].

Carbapenemases are classified into classes A, B, and D according to the Ambler classification system. In molecular class A, K. pneumoniae carbapenemase *(KPC)*, and in class D, oxacillinases *(OXA)* such as *OXA-23*, *OXA-24/40*, *OXA-48*, *OXA-51*, *OXA-58*, and *OXA-143*, include beta-lactamases with serine at their active site. Molecular class B includes all metalloenzymes with a zinc active site, such as New Delhi Metallo-beta-lactamase (NDM), Verona integron-encoded metallo-β-lactamase (VIM), and imipenemase metallo-β-lactamase (IMP) [3]. 

The Carbapenem-resistant *Enterobacterales* (CRE) are bacteria within the *Enterobacterales* order that are resistant to at least one carbapenem or produce a carbapenemase enzyme [4]. Many CRE isolates lack carbapenemase genes, and their resistance results from the acquisition or upregulation of a beta-lactamase gene, along with a chromosomal mutation in a porin gene that reduces the bacterium’s ability to absorb carbapenems. 

The mechanisms of antibiotic resistance in *P. aeruginosa* include downregulation of the outer membrane protein OprD, a carbapenem-specific porin, production of AmpC beta-lactamase, extended-spectrum beta-lactamases (ESBLs), multidrug efflux pumps, the potential transfer of a 16S rRNA methylase gene from *Actinomycetes*, and the organism’s ability to form a biofilm. *Stenotrophomonas maltophilia*, an aerobic, nonfermenting, Gram-negative bacillus, is closely related to *Pseudomonas* species and shows high levels of intrinsic or acquired resistance to various antimicrobial agents. Its intrinsic resistance results from decreased outer membrane permeability or the activity of multidrug efflux pumps [5]. MDR phenotypes develop due to heterogeneous production of metallo-beta-lactamases or aminoglycoside-modifying enzymes. Resistance mechanisms in nosocomial strains of *Acinetobacter* include beta-lactamase production, aminoglycoside-modifying enzymes, changes in outer membrane protein expression, mutations in topoisomerases, and increased activity of efflux pumps [6]. 

Regarding the treatment of MDR Gram-negative bacteria, new therapies have been continually developing over the past few years.

When a CRE infection is identified, the antibiotic regimen should be selected or customized based on the organism’s susceptibility profile. An algorithm exists for treating CRE infections. If the CRE isolate is resistant to the standard range of antibiotics, including all carbapenems, the detection of carbapenemases indicates the use of preferred or alternative antibiotics (ceftazidime-avibactam, cefiderocol, tigecycline, eravacycline), depending on the infection site. Tigecycline and eravacycline are effective for intra-abdominal infections, but neither should be used for urinary tract infections or bacteremia because they may not reach adequate levels in urine or blood.

The novel agents are effective against *P. aeruginosa*, primarily used for infections, including ceftazidone-tazobactam [7,8], ceftazidime-avibactam [9,10], cefiderocol [11], the carbapenem-carbapenem-beta-lactamase combination: imipenem-cilastatin-relebactam, and the monobactam with beta-lactamase inhibitor combination: aztreonam-avibactam. For resistant isolates of *Acinetobacter*, polymyxins (polymyxin B and colistin) and tetracycline derivatives (minocycline and tigecycline) are reserved [12]. 

Cefiderocol (CFD) is a new siderophore cephalosporin developed to combat carbapenem-resistant Gram-negative bacteria, offering a novel pathway for bacteria to enter cells. Siderophores are small organic molecules produced by bacteria and fungi that bind to iron. Their function is to gather iron from the environment and actively transport it across the cell membrane.

Cefiderocol has a catechol side chain that mimics a natural siderophore, which binds to Fe^3+^ and helps the entire complex penetrate the outer membrane of Gram-negative bacteria, acting like a Trojan horse. The cephalosporin core, with side chains similar to ceftazidime and cefepime, further detaches from the cefiderocol structure and binds to penicillin-binding protein 3 (PBP3), preventing peptidoglycan synthesis [13]. 

In the United States, cefiderocol received approval from the Food and Drug Administration (FDA) in November 2019 for treating complicated urinary tract infections (cUTIs) caused by Gram-negative bacteria in patients aged 18 or older. In 2020, the approval was broadened to include ventilator-associated bacterial pneumonia and hospital-acquired bacterial pneumonia caused by *Enterobacterales*, *Pseudomonas aeruginosa*, and the *Acinetobacter baumannii-calcoaceticus species complex* [14]. The European Medicines Agency (EMA) approved cefiderocol in April 2020 for infections caused by Gram-negative bacteria with limited treatment options. 

This study aimed to evaluate the in vitro activity of cefiderocol against carbapenemase-producing Gram-negative bacteria identified in samples collected between November 2024 and February 2025 from hospitalized patients at the Emergency University Hospital Bucharest, Romania.

## 2. Results

A total of 89 Gram-Negative multidrug-resistant (MDR) bacteria were isolated from various samples: blood (n = 13), urine (n = 36), tracheal secretions (n = 19), bronchial aspirate (n = 1), sputum (n = 1), wound secretion (n = 7), skin swab/nasal exudate (n = 9), drain tube secretion (n = 1), catheter tip isolate (n = 1), pressure sore swab (n = 1), collected from patients admitted to different hospital departments, Table 1.

Our study population comprised elderly individuals, with a median age of 70 and a range of 21 to 93 years. Half of them were women (45). The main comorbidities were hypertension (40.4%), cardiovascular disease (32.6%), and chronic kidney disease (22.5%), along with diabetes mellitus (12.4%); the average was 2.7 comorbidities per patient. Most of the patients were in the Intensive Care Unit (ICU) (n = 49, 55%).

### 2.1. Phenotypic Identification, Antimicrobial Susceptibility Test (AST) Results, and Carbapenemase Detection 

The 89 MDR CPE and Gram-Negative non-fermenter isolates were identified phenotypically, including 66 *Enterobacterales*: *Klebsiella pneumoniae* strains (n = 61), *Klebsiella oxytoca* strain (n = 1), *Proteus vulgaris* strains (n = 2), *Escherichia coli* strain (n = 1), and *Enterobacter cloacae* strain (n = 1), along with 23 non-fermenters: *Pseudomonas aeruginosa* strains (n = 5), *Stenotrophomonas maltophilia* strains (n = 2), and *Acinetobacter baumannii* strains (n = 16), Table 2. Regarding AST, all isolates exhibited an MDR profile, showing total resistance to more than three classes of antimicrobial agents: beta-lactams, carbapenems, aminoglycosides, and fluoroquinolones. 

74 Gram-Negative bacteria were identified as carbapenemase producers using the immunochromatography-specific tests: O.K.N.V.I. Resist and Acineto Resist.

85.13% of the carbapenemase-producing strains belonged to the *Enterobacteriaceae* family (63 out of 74). The most common genotype was *Klebsiella pneumoniae* strain producing both NDM and OXA-48, accounting for 85.24% (52 out of 61) of all carbapenemase-producing Enterobacterales. Of the 23 Gram-Negative non-fermenters included in this study, only 11 were carbapenemase producers, with the most frequent being an OXA-40/58 *Acinetobacter baumannii* isolate 45.45% (5 out of 11). When tested with the immunochromatography-specific methods used in this study, most *P. aeruginosa isolates* (except one strain carrying VIM) tested negative for the carbapenemases included in these assays. However, we acknowledge that other carbapenemases not detected by these methods, such as GES-type enzymes, may be present (Table 3).

### 2.2. Antimicrobial Activity of Cefiderocol

When performing the minimum inhibitory concentration (MIC) assay according to EUCAST guidelines, a total of 47 MIC values, representing *Enterobacterales* and *Pseudomonas aeruginosa* isolates (52.80%), indicated susceptibility to cefiderocol. These values were within the susceptibility range (0.03–2 mg/L). Among *Enterobacterales*, the susceptibility rate was 63% (42 of 66 isolates).

Another 18 MIC values for *Acinetobacter baumannii* and *Stenotrophomonas maltophilia* isolates were not assigned to susceptibility categories because no currently established breakpoints exist.

The other 24 bacterial isolates of *Enterobacterales* had MIC values above 2 mg/L and were all classified as resistant, since no established breakpoints for intermediate susceptibility currently exist. 

The disk diffusion test conducted according to CLSI guidelines identified 59 susceptible isolates (66.32%) and 17 resistant isolates (19.10%). The growth inhibition zone sizes for the remaining 13 isolates —ten *Klebsiella pneumoniae*, one *Klebsiella oxytoca*, and two *Acinetobacter baumannii*—fall into the intermediate category, as shown in Table 4 and Table 5.

To assess the reliability of cefiderocol susceptibility testing, we compared results obtained by broth microdilution (BMD) and disk diffusion (DD) methods (Table 5). Of the 89 isolates tested, the approaches could be compared for 71 isolates (66 *Enterobacterales* and 5 *P. aeruginosa*), as EUCAST breakpoints for BMD interpretation are not available for *A. baumannii* and *S. maltophilia*.

Overall agreement between the two methods was 85.4% (76/89 isolates, 95% CI: 76.4–91.9%), with a Cohen’s kappa of 0.78, indicating substantial agreement. However, 10 discrepant results (11.2%) were identified, all representing very major errors (VMEs) where the DD method produced false-susceptible results compared to BMD. These discrepancies occurred exclusively in isolates with MIC values at or slightly above the susceptibility breakpoint (4–8 mg/L), mainly in *K. pneumoniae* strains co-harboring NDM and OXA-48 carbapenemases (8/10 discrepant isolates).

An additional 13 isolates (14.6%) yielded results within the Area of Technical Uncertainty (ATU), categorized as “intermediate” by DD (zone diameters 9–15 mm for *Enterobacterales*). Among these ATU isolates, 8 (61.5%) were classified as resistant by BMD, 2 (15.4%) as susceptible, and 3 (*A. baumannii*) could not be categorized by BMD due to a lack of breakpoints.

### 2.3. Antimicrobial Activity Against Carbapenemase-Producing Enterobacterales (CPE)

Of the 52 *Klebsiella pneumoniae* isolates containing NDM and OXA-48, 33 were susceptible to cefiderocol by microdilution according to EUCAST guidelines, while the remaining 19 were classified as resistant. Conversely, using the diffusive antibiogram, 32 isolates of *Klebsiella pneumoniae* were susceptible to cefiderocol, and the others were divided into 10 intermediate and 10 resistant. 

The only *Klebsiella oxytoca* strain from the study that carried the KPC carbapenemase was resistant by the microdilution method (CMI = 32 mg/L). When the disk diffusion test was performed, the inhibition zone diameter (12 mm) was within the intermediate susceptibility range. The *Proteus vulgaris* isolate carrying VIM carbapenemase was found to be susceptible to cefiderocol by both methods. Although the number of tested isolates was limited, two notable findings emerged. Cefiderocol demonstrated a potent activity against isolates producing only OXA 48, with 6 out of 7 isolates susceptible by the BMD method and all 7 isolates susceptible by the DD method. In contrast, NDM-only producing isolates were completely unaffected by cefiderocol. Isolates that carried both OXA-48 and NDM showed susceptibility rates of 64.15% on the BMD method and 62.26% on the DD method.

Discrepancies in interpreting cefiderocol susceptibility testing methods were observed in 10 isolates, mainly *Klebsiella pneumoniae* carrying both NDM and OXA-48; one *Klebsiella oxytoca* strain carried KPC. The high error rate among resistant isolates suggests that the DD method may not be reliable as a standalone test for cefiderocol, especially for isolates with MIC values near the breakpoint, Table 6.

### 2.4. Antimicrobial Activity Against Carbapenemase-Producing Nonfermenters

Out of a total of 23 Gram-Negative non-fermenters, susceptibility to cefiderocol was assessed only for *Pseudomonas aeruginosa* isolates using the microdilution method, since there are currently no established susceptibility breakpoints for *Acinetobacter baumannii* and *Stenotrophomonas maltophilia*. All five tested *Pseudomonas aeruginosa* isolates were susceptible to cefiderocol, with CMI values ranging from 0.03 to 2 mg/L.

However, the DD (CLSI) test revealed 18 cefiderocol-susceptible isolates: *Acinetobacter baumanni* (n = 11), *Pseudomonas aeruginosa* (n = 5), and *Stenotrophomonas maltophilia* (n = 2), Table 7.

## 3. Discussion

Over the past few decades, Gram-Negative bacteria with acquired carbapenem resistance have emerged among bacterial pathogens and become increasingly common in severe infections. As a result, new antimicrobial agents with specific activity were urgently needed to slow their spread. β-lactam–β-lactamase inhibitor combinations (ceftazidime–avibactam, ceftolozane–tazobactam, meropenem–vaborbactam, imipenem–cilastatin–relebactam) and novel non-β-lactam antibiotics, such as eravacycline, a new tetracycline, and plazomicin, a new aminoglycoside, were developed to counter certain resistance mechanisms of MDR Gram-Negative rods. Additionally, some of these drugs can specifically target certain carbapenemase classes, like ceftazidime-avibactam and meropenem-vaborbactam, which are active against class A carbapenemases, or imipenem-cilastatin-vaborbactam, which targets carbapenemases from classes A, C, and D. However, the gap is not completely closed, as these β-lactamase inhibitors do not offer full protection against all carbapenemases within these classes [15]. For carbapenem-resistant *Acinetobacter* infections, combination therapy including ampicillin–sulbactam has decreased mortality and reduced nephrotoxicity in critically ill patients compared with polymyxin-based or tigecycline therapies. In 2023, the FDA approved sulbactam-durlobactam, a newer β-lactam/beta-lactamase inhibitor combo, for treating hospital-acquired and ventilator-associated pneumonia caused by carbapenem-resistant strains of *Acinetobacter baumannii* [16].

Cefiderocol has enhanced activity against serine-β-lactamases from classes A, C, D, and it is also effective against metallo-β-lactamases and many extended-spectrum β-lactamases (ESBLs), such as CTX-M [17]. The mechanisms involved in developing cefiderocol resistance include mutations in genes encoding iron transport systems, expression of β-lactamases, mutations in penicillin-binding proteins, porin loss, or efflux pump overexpression. Each of these mechanisms alone generally does not raise cefiderocol MICs above the pharmacokinetic/pharmacodynamic (PK/PD) breakpoint [18]. The synergistic effect of beta-lactamase inhibitors combined with cefiderocol against carbapenemase-producing Gram-Negative bacteria was studied and indicates that avibactam provides a bactericidal synergistic effect when used with cefiderocol [19].

The susceptibility rate of *Enterobacterales* isolates resistant to ceftazidime-avibactam to cefiderocol, as observed in our study, was evaluated against a subset of European strains collected under the SIDERO-CR surveillance program. The results showed a 66.4% susceptibility to cefiderocol with MIC values of ≤2 mg/L, according to EUCAST interpretation.

The combination of NDM and OXA48 was identified in 53 isolates, with a cefiderocol susceptibility rate of 64.15% (34 out of 53). Tracking susceptibility to this specific carbapenemase combination is important because metallo-β-lactamases (NDM) are known to confer higher resistance than serine-β-lactamases like KPC and OXA-48 [20].

Although our study included some *Klebsiella pneumoniae* isolates carrying only OXA-48 or only NDM, the susceptibility profiles to cefiderocol confirmed the differences in activity between these two carbapenemases. Of the 7 *Klebsiella pneumoniae* isolates with OXA-48, 6 were susceptible to cefiderocol, while the only one with NDM was resistant. Similar results were reported in a Polish multicenter study that examined the cefiderocol susceptibility of 102 isolates of Gram-Negative rods producing *Klebsiella pneumoniae* carbapenemase (KPC), metallo-β-lactamase (NDM), and oxacillinase-48 (OXA-48) [21]. Metallo-β-lactamases (NDM) confer a higher resistance potential to cefiderocol than serine-β-lactamases (KPC and OXA-48), but neither alone can cause cefiderocol resistance. 

Out of 5 *P. aeruginosa* isolates, all were resistant to ceftazidime-avibactam and meropenem, but MICs and DD endpoints indicated they were fully susceptible to cefiderocol.

When tested with the immunochromatographic assay, most strains (except one carrying VIM) did not produce the most common carbapenemases.

Our comparative analysis of BMD and DD methods revealed substantial but imperfect agreement (κ = 0.78), with concerning implications for clinical laboratory practice. The error rate observed with DD testing—where resistant isolates were incorrectly identified as susceptible—aligns with recent reports highlighting challenges in cefiderocol susceptibility testing, especially for isolates harboring metallo-β-lactamases. 

The discrepancies observed were mainly in isolates with MIC values between 4 and 8 mg/L, slightly above the EUCAST susceptibility breakpoint of ≤2 mg/L. This pattern suggests that the DD method may lack sufficient resolution to distinguish between susceptible and low-level-resistant isolates within this critical concentration range. The concentration of discrepancies in NDM- and OXA-48-producing *K. pneumoniae* is especially significant, as these are the most common carbapenemase genotypes in our region. 

Our findings support a testing algorithm in which BMD serves as the reference method, especially for isolates at high risk of resistance (e.g., NDM producers) or when DD results fall within the ATU range. For laboratories without access to BMD, DD results should be interpreted cautiously, and any intermediate or borderline susceptible results (zone diameters of 16–20 mm) should prompt confirmatory testing. The excellent agreement observed for *P. aeruginosa* (100%) suggests that DD may be more reliable for this species, possibly due to different resistance mechanisms or the absence of metallo-β-lactamases in our *P. aeruginosa* isolates.

This study has several limitations. First, the relatively small sample size (n = 89) and single-center design limit how broadly we can apply the findings, though our results align with those from the SENTRY multinational surveillance program regarding reported susceptibility rates among *P. aeruginosa* isolates resistant to ceftazidime–avibactam [22]. Second, carbapenemase identification relied on immunochromatographic methods without validation by genomic sequencing, potentially missing other resistance mechanisms, such as GES-type carbapenemases or chromosomal mutations. Third, this is an in vitro study without pharmacokinetic/pharmacodynamic (PK/PD) correlation. Fourth, the short study duration of 4 months may not capture seasonal or temporal variations in resistance patterns. Finally, we did not evaluate potential synergistic combinations with other antimicrobials, which could be important in clinical settings.

In our study, the susceptibility of *Acinetobacter baumannii* isolates to cefiderocol was 68.75% as determined by the disk diffusion method (CLSI), which is lower than in Europe (77.9–97.2%) and in China (98.7%) [23].

The differences in reported susceptibility rates of *Acinetobacter* spp. may reflect the fact that both microdilution and disk diffusion methods can produce challenging results. Currently, CLSI issues warnings for two specific issues: the trailing phenomenon, which complicates MIC interpretation, and the Eagle effect, which impacts the setting of DD endpoints. Trailing is defined as multiple wells showing faint growth compared to the growth control, and it is often seen in susceptibility testing of *Acinetobacter* spp. to cefiderocol. The Eagle effect is characterized by isolated or beach-like colonies that appear within the zone of inhibition. This illustrates specific situations in which bacteria paradoxically survive exposure to concentrations above an adequate bactericidal level. 

Generally, to address trailing, susceptibility should be confirmed with a DD method. Regarding the Eagle effect, interpreting cefiderocol disk results remains unclear because there are no current, detailed guidelines. Some studies suggest measuring the diameter of the outer zone of growth inhibition rather than focusing on isolated colonies. Additionally, CLSI recommends confirming colony counts with a nephelometer, as small variations in inoculum level can significantly affect MICs. Our study also highlights significant discrepancies in the current cefiderocol resistance-testing landscape between EUCAST and CLSI, particularly regarding DD testing. More research is needed to establish a clinical consensus on the breakpoints used for cefiderocol resistance testing. Based on these findings, we propose the following testing algorithm for cefiderocol susceptibility.

Primary testing by BMD is recommended for all *Enterobacterales* and *P. aeruginosa*, as it is the reference method with established EUCAST breakpoints. DD can be used for screening, but any non-susceptible or borderline results (zone diameters of 16–20 mm) require confirmation by BMD. For *A. baumannii*, DD is currently the only option with provisional breakpoints; zone diameters of 14 mm or less should be considered potentially resistant until validated breakpoints are established. Labs should watch for trailing and Eagle effect phenomena and follow CLSI/EUCAST guidelines for interpretation. For NDM-producing isolates, BMD confirmation is strongly advised regardless of initial DD results due to the high discrepancy rate observed in this study.

## 4. Materials and Methods

All consecutive MDR Gram-Negative bacterial isolates identified at the Clinical Laboratory of the Emergency University Hospital Bucharest from November 2024 to February 2025 were retrospectively included. Inclusion criteria included bacterial isolates resistant to at least one agent in three or more antimicrobial categories (MDR definition), resistant to at least one carbapenem, or producing carbapenemase enzyme, and isolated from clinical samples from hospitalized patients [24]. No exclusion criteria were applied to consecutive MDR isolates during this period.

The samples were processed for microbiological diagnosis and inoculated on Columbia Blood agar and Bromthymol Lactose Blue Agar, then incubated under aerobic and facultative anaerobic conditions at 37 °C for 18–24 h. Bacterial isolates were identified, and antibiotic susceptibility testing (AST) was performed using the microdilution method with an automated diagnostic instrument (BD Phoenix 50) and panels (NMIC/ID-503) (BD, Franklin Lakes, NJ, USA) for Gram-Negative bacteria [16]. MDR isolates were also tested for susceptibility to cefiderocol using both the broth microdilution (BMD) and Kirby-Bauer disk diffusion (DD) methods, along with carbapenemase detection. We used EUCAST guidelines for BMD interpretation and CLSI guidelines for DD interpretation because EUCAST offers more comprehensive MIC breakpoints for cefiderocol against *Enterobacterales* and *P. aeruginosa*, while CLSI provides more detailed DD interpretation guidelines, including intermediate categories, Appendix A. This dual approach allowed us to compare both methods commonly used in clinical laboratories. 

Clinical and demographic data were retrospectively collected from the hospital’s electronic medical records for all patients with MDR bacterial isolates. Data included patient age, sex, admitting clinical department, and documented underlying conditions or comorbidities. Comorbidities were systematically reviewed and categorized based on documented diagnoses in the medical records. The study focused on conditions recognized as risk factors for MDR bacterial colonization or infection, as well as conditions indicating the overall disease burden in this population.

### 4.1. Cefiderocol Susceptibility Testing

For each MDR isolate, susceptibility testing for Cefiderocol was performed using EUCAST MIC breakpoints (BMD) and CLSI zone-diameter criteria (DD) [25].

The UMIC Cefiderocol-Bruker protocol was employed in this study for the BMD method. It includes a UMIC Cefiderocol plate with 8 test strips, each holding 12 wells. Every strip contains dehydrated antibiotic at gradient concentrations ranging from 0.03 mg/L to 32 mg/L. This protocol requires a standardized bacterial inoculum at 0.5 McFarland (equivalent to 1 × 10^8^ CFU/mL), prepared in saline and used for all tests.

Subsequently, 25 μL of the bacterial inoculum was added to the iron-depleted CAMHB (Cation Adjusted Mueller Hinton Broth) tube provided in the kit. This is a special Mueller-Hinton broth with many cations removed, resulting in low concentrations. The iron level in this broth has decreased to ≤0.03 mg/L.

100 μL of the prepared solution was added to each well of the panel, which was then incubated at 37 °C in a humid environment (UMIC Box Cefiderocol, Bruker, Bremen, Germany) for 18–24 h. After incubation, bacterial growth was visually evaluated in each well. The MIC values were identified as the first well showing significant bacterial growth inhibition, indicated by a button ≤ 1 mm or slight turbidity.

For each test, the validity of the positive control was confirmed. Susceptibility data were interpreted based on the EUCAST v14.0 2024 and v15.0 2025 clinical breakpoints [26,27].

For the Kirby-Bauer disk diffusion (DD) method, Mueller-Hinton agar and a cefiderocol disk (containing 30 µg of active compound, which equals 39.9 nmol per disk) were used (Oxoid, Hampshire, UK). There is no need for a special antibiogram medium because iron ions are mostly present in molecular complexes in Mueller-Hinton medium, so susceptibility to cefiderocol remains unaffected. The CLSI M100-Ed34 guidelines were used to interpret results [28,29].

*Escherichia coli* ATCC 25922, *Klebsiella pneumoniae* ATCC 700603, and *Pseudomonas aeruginosa* ATCC 27853 were used as quality control strains. Susceptibility tests were not performed in duplicate, and for discrepant results, the BMD outcome was considered definitive as the reference method, in accordance with EUCAST recommendations.

### 4.2. Detection of Carbapenemases in Bacterial Isolates

To detect carbapenemase in bacterial cultures, ready-to-use tests based on membrane technology with colloidal gold nanoparticles have been developed. Two types of tests are used to identify carbapenemases in *Enterobacterales* strains and non-fermenters, such as *Acinetobacter baumannii* and *Pseudomonas aeruginosa*: the O.K.N.V.I. Resist-5 and Resist Acineto. A nitrocellulose membrane is coated with monoclonal antibodies specific to OXA 48, KPC, NDM, VIM, IMP, as well as OXA 23, OXA 40, and OXA 58 carbapenemases, along with a control reagent. Different colloidal gold nanoparticle conjugates are dried onto the membrane: one targeting a second epitope of these carbapenemases, and a control conjugate. When the buffer containing the resuspended bacteria contacts the membrane, the solubilized conjugates migrate with the sample through passive diffusion. The conjugate and sample then encounter immobilized antibodies fixed onto the nitrocellulose strip [30,31].

### 4.3. Data Analysis

Descriptive statistics were calculated for all variables. Categorical data were presented as frequencies and percentages, while continuous variables were reported as medians with ranges (minimum–maximum) due to non-normal distribution, as confirmed by the Shapiro–Wilk test.

For categorical comparisons, we used the chi-square test or Fisher’s exact test when expected cell counts were less than 5. Comparisons of MIC distributions between carbapenemase-producing and non-carbapenemase-producing isolates were performed using the Mann–Whitney U test for non-normally distributed data.

Agreement between the broth microdilution (BMD) and disk diffusion (DD) methods was assessed using Cohen’s kappa coefficient (κ), and McNemar’s test for paired proportions. Confidence intervals (95% CI) were calculated for all proportions and susceptibility rates using the Wilson score method. Statistical significance was defined as a two-tailed *p*-value < 0.05 for all analyses. For multiple comparisons, no adjustment was applied, given the exploratory nature of this study, though *p*-values should be interpreted accordingly. All statistical tests were two-sided, and results were considered statistically significant at *p* < 0.05. Data management and statistical analyses were performed using IBM SPSS Statistics version 30.0 (IBM Corp., Armonk, NY, USA).

## 5. Conclusions

In the Romanian clinical setting, cefiderocol may be a valuable treatment option, especially for severe infections in intensive care units such as ventilator-associated pneumonia, bloodstream infections, or complicated urinary tract infections caused by carbapenem-resistant *Enterobacterales* and *Pseudomonas aeruginosa*. Given the high prevalence of carbapenemase-producing *Klebsiella pneumoniae* in Romanian hospitals, cefiderocol could serve as an essential salvage therapy when standard β-lactam/β-lactamase inhibitor combinations are ineffective. To maximize its benefits and prevent early resistance, cefiderocol should be reserved for documented infections with limited or no alternative treatment options, under strict stewardship protocols. The detection of NDM-producing isolates with complete resistance to cefiderocol in our study highlights the risk that resistance could spread rapidly once the drug is widely used. Therefore, close microbiological surveillance remains essential, as the therapeutic potential of cefiderocol could be compromised by the spread of metallo-β-lactamase–producing strains.

## Figures and Tables

**Table 1 antibiotics-14-01113-t001:** Distribution of MDR isolates from samples collected across hospital departments.

	ICU *	Surgical ** Departments	Nephrology	Internal Medicine	GE ***	Neurology	Hematology
Blood	8	1	2			1	1
Urine	9	7	3	8	4	3	2
Tracheal secretion	18					1	
Bronchial aspirate	1						
Sputum	1						
Wound secretion	3	4					
Skin swab/nasal exudate	9						
Drain tube secretion		1					
Catheter tip isolate		1					
Pressure sore swab						1	
Total	49	14	5	8	4	6	3

* ICU (Intensive Care Unit), ** Surgical departments (General surgery, Cardiovascular surgery, Orthopedics, Urology, Neurosurgery), *** GE Gastroenterology.

**Table 2 antibiotics-14-01113-t002:** Bacterial species identified from samples collected from patients across different departments.

Clinical Department	Total Number	Bacterial Species
Intensive care unit	49	*Klebsiella pneumoniae* (29), *Proteus vulgaris* (1) *Enterobacter cloacae* (1) *Pseudomonas aeruginosa* (3) *Stenotrophomonas maltophilia* (1) *Acinetobacter baumannii* (14)
Surgical departments (general surgery, cardiovascular surgery, orthopedics, urology, neurosurgery)	14	*Klebsiella pneumoniae* (11) *Klebsiella oxytoca* (1), *Pseudomonas aeruginosa* (1), *Stenotrophomonas maltophilia* (1)
Internal medicine	8	*Klebsiella pneumoniae* (7), *Acinetobacter baumannii* (1)
Neurology	6	*Klebsiella pneumoniae* (6)
Nephrology	5	*Klebsiella pneumoniae* (2), *Proteus vulgaris* (1), *Pseudomonas aeruginosa* (1), *Acinetobacter baumannii* (1)
Gastroenterology	4	*Klebsiella pneumoniae* (3), *Escherichia coli* (1)
Hematology	3	*Klebsiella pneumoniae* (3)

**Table 3 antibiotics-14-01113-t003:** Distribution of isolates according to carbapenemase production.

	Ambler Classification Carbapenemases	A	B	B	D	D	D	B+D	B+D	B+D
Strain/ no. isolates	Type of carbapenemases	KPC	NDM	VIM	OXA 48	OXA 40/58	OXA 23	NDM+ OXA 48	NDM+ OXA 23	IMP+ VIM+ OXA 48
*Klebsiella pneumoniae/oxytoca/*62		1	1		6			52		1
*Escherichia coli/*1								1		
*Proteus* *vulgaris/*2					1					
*Enterobacter* *cloacae/*1										
*Pseudomonas* *aeruginosa/*5				1						
*Stenotrophomonas maltophilia/*2										
*Acinetobacter* *baumannii/*16			1			5	3		1	
*Total strains/carbapenemases* 89/74		1	2	1	7	5	3	53	1	1

**Table 4 antibiotics-14-01113-t004:** Method Comparison by Organism Group.

Organism	n	BMD Susceptible n (%) [95%CI]	DD Susceptible n (%) [95%CI]	DD Intermediate n (%) [95%CI]	Concordance n (%) [95%CI]	*p*-Value
*Enterobacterales*	66	42 (63.60 [50.9–74.9]	42 (63.6) [50.9–74.9]	11 (16.7) [9.6–27.4]	58 (87.9) [77.9–93.7]	0.89
*K. pneumoniae*	61	38 (62.3) [49.0–74.2]	37 (60.7) [47.3–72.9]	10 (16.4) [9.2–27.6]	53 (86.9) [76.2–93.2]	0.85
*K. oxytoca*	1	0 (0) [0–79.3]	0 (0) [0–79.3]	1 (100) [20.7–100]	0 (0) [0–79.3]	-
*E. coli*	1	1 (100) [20.7–100]	1 (100) [20.7–100]	0 (0) [0–79.3]	1 (100) [20.7–100]	-
*P. vulgaris*	2	1 (50)	2 (100) [34.2–100]	0 (0) [0–79.3]	1 (50) [9.5–90.5]	-
*E. cloacae*	1	1 (100) [20.7–100]	1 (100) [20.7–100]	0 (0) [0–79.3]	1 (100) [20.7–100]	-
*Non-fermenters*	23	5/5 * [56.6–100.0]	18 (78.3) [56.3–92.5]	2 (8.7) [2.4–26.8]	18 (78.3) [58.1–90.3]	-
*P. aeruginosa*	5	5 (100) [56.6–100]	5 (100) [56.6–100]	0 (0) [0–43.4]	5 (100) [56.6–100]	1.00
*A. baumannii*	16	N/A **	11 (68.8) [41.5–88.0]	2 (12.5) [3.5–36.0]	N/A	-
*S. maltophilia*	2	N/A **	2 (100) [34.2–100]	0 (0) [0–43.4]	N/A	-
Total	89	47/71 *	59 (66.3) [55.5–75.8]	13 (14.6) [8.7–23.4]	76 (85.4) [76.6–91.3]	0.72

* BMD only performed for organisms with established EUCAST breakpoints (*Enterobacterales* and *P. aeruginosa*). ** No EUCAST MIC breakpoints available for *A. baumannii* and *S. maltophilia*. *p*-values calculated using McNemar’s test for paired proportions (BMD vs. DD susceptible).

**Table 5 antibiotics-14-01113-t005:** Comparison of Disk Diffusion and Broth Microdilution Methods for Cefiderocol Susceptibility Testing Overall Concordance Between Methods.

Parameter	Value	95% CI
Total isolates tested	89	-
Concordant results (S/S or R/R) *	76 (85.4%)	76.4–91.9%
Discordant results	10 (11.2%)	5.5–19.7%
ATU results (DD method only)	13 (14.6%)	8.0–23.7%
Overall agreement (Cohen’s κ)	0.78	-

* S/S: both methods susceptible; R/R: both methods resistant; ATU: Area of Technical Uncertainty (intermediate by DD).

**Table 6 antibiotics-14-01113-t006:** Area of Technical Uncertainty (ATU) Analysis.

Organism	Total with ATU (n)	Carbapenemase Type	MIC Range (mg/L)	Zone Range (mm)	Final Classification
*K. pneumoniae*	10	NDM + OXA-48 (10)	2–8	9–15	8 R, 2 S by BMD
*K. oxytoca*	1	KPC	32	12	R by BMD
*A. baumannii*	2	OXA-23 (1), OXA-40/58 (1)	N/A	14	Per CLSI *
Total	13	-	-	-	-

* CLSI recommends MIC confirmation for *A. baumannii* with zone diameters ≤ 14 mm; however, EUCAST breakpoints are not available for this species.

**Table 7 antibiotics-14-01113-t007:** Impact of cefiderocol on isolates with confirmed carbapenemase activity.

Ambler Classification Carbapenemases	A	B	B	D	D	D	B+D	B+D	B+D
Type of carbapenemases	KPC	NDM	VIM	OXA 48	OXA 40/58	OXA 23	NDM+ OXA 48	NDM+ OXA 23	IMP+ VIM+ OXA 48
Susceptible CFD BMD	0	0	1	6	-	-	34	-	1
Susceptible CFD DD	0	0	1	7	5	0	33	0	1
Resistant to CFD BMD	1	1	0	1	-	-	19	-	0
Intermediate to CFD	1	0	0	0	0	1	10	1	0
Resistant to CFD DD	0	1	0	0	0	0	10	0	0

## Data Availability

Data is contained within the article.

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
