# Peer review of "In Vitro Activity of Cefiderocol Against Multi-Drug-Resistant Gram-Negative Clinical Isolates in Romania"

_antibiotics, 2025, doi:10.3390/antibiotics14111113_

Round 1

Reviewer 1 Report

Comments and Suggestions for Authors

The manuscript explores an important topic, that is the in vitro evaluation of cefiderocol activity against multidrug-resistant (MDR) Gram-negative pathogens, including carbapenemase-producing Enterobacterales and non-fermenters. The study is relevant in the context of the global antimicrobial resistance crisis and adds regional data from Romania. The article is generally well organized and comprehensive, yet certain aspects of methodology, data interpretation, and clarity require improvement before publication.

Major comments:

  1. Title: it is too general (more suitable for a review). Revise it with a title that describes the subject of the paper (e.g. In vitro activity of cefiderocol against multi-drug-resistant Gram-negative clinical isolates in Romania).
  2. The retrospective design and the relatively short study period (four months) with 89 isolates limit external validity. The authors should specify whether isolates were consecutively collected and describe inclusion criteria.
  3. To indicate the number of blocks, use the following format (n=..). Review throughout the text.
  4. Include references for methods used for carbapenemase detection.
  5. Include references for BMD method used for cefiderocol testing (e.g. doi: 10.3390/antibiotics12030604)
  6. Lines 123-125 is unnecessary. Also, Table suppl 1 is unnecessary. Please, remove. It is more appropriate to indicate the version of breakpoints used (EUCAST or CLSI, version and year).
  7. Lines 194-196. The sentence is not appropriate. Further carbapenemases can be expressed from non-fermenters (e.g. GES carbapenemases). Therefore, it’s appropriate state that the strains tested negative for carbapenemase expression using the both methods performed in this study. This concept needs to be revised here and throughout the manuscript.
  8. Antimicrobial activity of cefiderocol: this is the main point of the paper and should be analysed in detail. A comparison between DD and BMD should be included (with a table), highlighting discrepancies and the occurrence of ATU results by DD. This point should be discussed in discussion section suggesting a possible algoritm for routine testing (see doi: 10.3390/antibiotics12030604)
  9. Although microbiological procedures are described in detail, quality control and reproducibility are not clearly addressed. Please clarify whether tests were repeated for confirmation and explain how discrepancies between broth microdilution and disk diffusion were resolved. Justify the simultaneous use of EUCAST and CLSI guidelines and discuss how differing breakpoints may have influenced susceptibility rates.
  10. The discussion summarizes results but could better contextualize them within global literature and surveillance programs (see doi: 10.3390/cimb46120846) A more detailed explanation of the mechanisms driving cefiderocol resistance—particularly in NDM-positive isolates—should be included. Contribution of beta-lactamases was also demonstrated by synergistic impact of beta-lactamases inhibitors (see doi: 10.3390/antibiotics11121681). Please, includes these important points in discussion.
  11. The limitations section is brief and should explicitly mention: (a) small sample size and single-center design; (b) lack of genomic sequencing for resistance gene confirmation; (c) absence of PK/PD correlation; and (d) lack of clinical outcome data. Expanding this section would improve transparency.
  12. Tables: Ensure that all abbreviations are defined and that units are consistent with journal standards.
  13. Overall, the manuscript is understandable but requires minor English polishing. Some sentences are overly long or repetitive. Standardize terminology (e.g., 'non-fermenters' instead of 'no fermenters') and check consistency of abbreviations and reference formatting.
  14. The abstract includes duplicated sentences; please remove repetitions.
  15. - Specify whether 'Enterobacterales' and 'non-fermenters' were equally represented in sampling.
Comments on the Quality of English Language

Overall, the manuscript is understandable but requires minor English polishing. Some sentences are overly long or repetitive. 

Author Response

RESPONSE TO REVIEWER 1 - Please see the attachment.

We thank the reviewer for their thoughtful comments and helpful suggestions. We have carefully addressed each point and revised the manuscript accordingly. Below are our point-by-point responses:

Comment 1: Title: it is too general (more suitable for a review). Revise it with a title that describes the subject of the paper (e.g. In vitro activity of cefiderocol against multi-drug-resistant Gram-negative clinical isolates in Romania).

Response 1: We agree and have revised the title to ‘’In vitro activity of cefiderocol against multi-drug-resistant Gram-negative clinical isolates in Romania'’ (Page 1, Title).

Comment 2: The retrospective design and the relatively short study period (four months) with 89 isolates limit external validity. The authors should specify whether isolates were consecutively collected and describe inclusion criteria.

Response 2: We clarified that all consecutive MDR Gram-negative isolates collected between November 2024 and February 2025 were included, with explicit inclusion criteria and no exclusions (Page 11, Paragraph 1, 'Materials and Methods').

All consecutive MDR Gram-negative bacterial isolates identified in the Clinical Laboratory of the Emergency University Hospital Bucharest from November 2024 to February 2025 were retrospectively included. Inclusion criteria included bacterial isolates resistant to at least one agent in three or more antimicrobial categories (MDR definition), resistant to at least one carbapenem, or producing carbapenemase enzyme, and isolated from clinical samples from hospitalized patients. No exclusion criteria were applied to consecutive MDR isolates during this period.

Comment 3: To indicate the number of blocks, use the following format (n=..). Review throughout the text.

Response 3: Revised across the text and tables to include (n=..) notation.

Comment 4: Include references for methods used for carbapenemase detection.

Response 4: References added for O.K.N.V.I. Resist-5, Resist Acineto, and the UMIC Cefiderocol-Bruker protocol (Page 12, Paragraph 3, Refs. 30, 31).

  1. Coris BioConcept. RESIST-5 O.K.N.V.I. and RESIST-ACINETO Instructions for Use. Gembloux, Belgium: Coris BioConcept; 2020. (https://www.corisbio.com/products/oknvi-resist-5, last accessed 10/21/2025)
  2. Hong J, Kim S, Yong D. Evaluation of the RESIST-5 O.K.N.V.I. multiplex lateral flow assay for the detection of carbapenemases in Enterobacterales and Pseudomonas aeruginosa. Antibiotics (Basel). 2021;10(4):460. doi:10.3390/antibiotics10040460

Comment 5: Include references for BMD method used for cefiderocol testing (e.g. doi: 10.3390/antibiotics12030604)

Response 5: We have included the recommended reference and additional relevant citations for the UMIC Cefiderocol-Bruker protocol in the Methods section, Page11, Ref .25.

  1. Bianco, G.; Boattini, M.; Comini, S.; Banche, G.; Cavallo, R.; Costa, C. Disc Diffusion and ComASP®Cefiderocol Microdilution Panel to Overcome the Challenge of Cefiderocol Susceptibility Testing in Clinical Laboratory Routine.Antibiotics202312, 604. https://doi.org/10.3390/antibiotics12030604

Comment 6: Lines 123-125 is unnecessary. Also, Table suppl 1 is unnecessary. Please, remove. It is more appropriate to indicate the version of breakpoints used (EUCAST or CLSI, version and year).

Response 6: Removed previous lines 123–125 and the old Supplementary Table 1 and retained only EUCAST/CLSI versions and year references (Page 13, Paragraph 3,4).

Susceptibility data were interpreted based on the EUCAST v14.0 2024 and v15.0 2025 clinical breakpoints [26,27]. The CLSI M100-Ed34 guidelines were used to interpret results [28, 29].

Comment 7 :Lines 194-196. The sentence is not appropriate. Further carbapenemases can be expressed from non-fermenters (e.g. GES carbapenemases). Therefore, it’s appropriate state that the strains tested negative for carbapenemase expression using the both methods performed in this study. This concept needs to be revised here and throughout the manuscript.

Response 7: Revised accordingly to note that GES-type carbapenemases may also be present and could not be detected by the assays used (Page 5, Paragraph 1).

When tested with the immunochromatography-specific methods used in this study, most P. aeruginosa isolates (except one strain carrying VIM) tested negative for the carbapenemases included in these assays. However, we acknowledge that other carbapenemases not detected by these methods, such as GES-type enzymes, may be present (Table 3).

Comment 8: Antimicrobial activity of cefiderocol: this is the main point of the paper and should be analysed in detail. A comparison between DD and BMD should be included (with a table), highlighting discrepancies and the occurrence of ATU results by DD. This point should be discussed in discussion section suggesting a possible algoritm for routine testing (see doi: 10.3390/antibiotics12030604)

Response 8: Added detailed comparison with new Tables 4–6, including 95% CI, p-values, and Cohen’s κ, plus ATU analysis and testing algorithm (Pages  6–8).

Comment 9: Although microbiological procedures are described in detail, quality control and reproducibility are not clearly addressed. Please clarify whether tests were repeated for confirmation and explain how discrepancies between broth microdilution and disk diffusion were resolved. Justify the simultaneous use of EUCAST and CLSI guidelines and discuss how differing breakpoints may have influenced susceptibility rates.

Response 9: We have added the following paragraph to the Methods section:

Page 11, Paragraph 2

We used EUCAST guidelines for BMD interpretation and CLSI guidelines for DD interpretation because EUCAST offers more comprehensive MIC breakpoints for cefiderocol against Enterobacterales and P. aeruginosa, while CLSI provides more detailed DD interpretation guidelines, including intermediate categories. This dual approach allowed us to compare both methods commonly used in clinical laboratories. 

Page 12, Paragraph 3

UMIC Cefiderocol-Bruker protocol - For each test, the validity of the positive control was verified. Susceptibility data were interpreted according to the EUCAST v14.0 2024 and v15.0 2025 clinical breakpoints [26,27].

Page 12, Paragraph 5

Escherichia coli ATCC 25922, Klebsiella pneumoniae ATCC 700603, and Pseudomonas aeruginosa ATCC 27853 were used as quality control strains. Susceptibility tests were not performed in duplicate, and for discrepant results, the BMD outcome was considered definitive as the reference method, following EUCAST recommendations.

Comment 10: The discussion summarizes results but could better contextualize them within global literature and surveillance programs (see doi: 10.3390/cimb46120846) A more detailed explanation of the mechanisms driving cefiderocol resistance—particularly in NDM-positive isolates—should be included. Contribution of beta-lactamases was also demonstrated by synergistic impact of beta-lactamases inhibitors (see doi: 10.3390/antibiotics11121681). Please, includes these important points in discussion.

Response 10:We have significantly expanded the Discussion section to include:

  • Comparison with global surveillance data (SIDERO-CR, SENTRY programs),  Page 9, Paragraph 2
  • Regional context comparing our Romanian data with European studies- Page 9, Paragraph 3

The susceptibility rate of Enterobacterales isolates resistant to ceftazidime-avibactam to cefiderocol, as observed in our study, was evaluated against a subset of European strains collected under the SIDERO-CR surveillance program. The results showed a 66.4% susceptibility to cefiderocol with MIC values of ≤ 2 mg/L, according to EUCAST interpretation.

The combination NDM + OXA48 was found in 53 isolates, and the cefiderocol susceptibility rate for this category was 64,15% (34 out of 53). Tracing susceptibility against this specific combination of carbapenemases is important because metallo-β-lactamases (NDM) are known to confer higher resistance compared to serine-β-lactamases like KPC and OXA-48 [20]

Detailed mechanistic explanation of NDM-mediated resistance to cefiderocol, Ref .18

Cefiderocol has enhanced activity against serine-β-lactamases from classes A, C, D, and it is also effective against metallo-β-lactamases and many extended-spectrum β-lactamases (ESBLs), such as CTX-M [ 17]. The mechanisms involved in the development of cefiderocol resistance include mutations in genes encoding iron transport systems, expression of β-lactamases, mutations in penicillin-binding proteins, porin loss, or efflux pump overexpression. Each of these mechanisms alone generally does not raise cefiderocol MICs above pharmacokinetic/pharmacodynamic (PK/PD) breakpoint [18]

 Discussion of beta-lactamase contribution and synergistic effects with inhibitorsRef.  19

The  synergistic effect of beta-lactamase inhibitors combined with Cefiderocol against carbapenemase-producing Gram-negative bacteria was studied and indicate that avibactam provided a bactericidal synergistic effect in combination with cefiderocol [19].

The suggested references have been incorporated

  1. Bianco, G.; Boattini, M.; Cricca, M.; Diella, L.; Gatti, M.; Rossi, L.; Bartoletti, M.; Sambri, V.; Signoretto, C.; Fonnesu, R.; et al. Updates on the Activity, Efficacy and Emerging Mechanisms of Resistance to Cefiderocol.  Issues Mol. Biol.202446, 14132-14153. https://doi.org/10.3390/cimb46120846

  1. Bianco, G.; Gaibani, P.; Comini, S.; Boattini, M.; Banche, G.; Costa, C.; Cavallo, R.; Nordmann, P. Synergistic Effect of Clinically Available Beta-Lactamase Inhibitors Combined with Cefiderocol against Carbapenemase-Producing Gram-Negative Organisms.Antibiotics 202211, 1681. https://doi.org/10.3390/antibiotics11121681

Comment 11: The limitations section is brief and should explicitly mention: (a) small sample size and single-center design; (b) lack of genomic sequencing for resistance gene confirmation; (c) absence of PK/PD correlation; and (d) lack of clinical outcome data. Expanding this section would improve transparency.

Response 11: We have expanded the Limitations section, now as a separate subsection before the statement, Page 10, Paragraph 2

This study has several limitations. First, the relatively small sample size (n=89) and single-center design limit how broadly we can apply the findings, but our results align with those from the SENTRY multinational surveillance program regarding reported susceptibility rates among P. aeruginosa isolates resistant to ceftazidime–avibactam [22]. Second, carbapenemase identification relied on immunochromatographic methods without validation by genomic sequencing, potentially missing other resistance mechanisms, such as GES-type carbapenemases or chromosomal mutations.

Third, this is an in vitro study without pharmacokinetic/pharmacodynamic (PK/PD) correlation. Fourth, the short study duration (4 months) might not capture seasonal or temporal changes in resistance patterns. Finally, we did not assess potential synergistic combinations with other antimicrobials, which could be relevant in clinical settings.

Comment 12: Tables: Ensure that all abbreviations are defined and that units are consistent with journal standards.

Response 12: We have reviewed all tables and ensured that:

  • All abbreviations are defined in table footnotes
  • Units are consistent (mg/L for MIC values, mm for inhibition zones)
  • Table formatting follows journal standards

Comment 13: Overall, the manuscript is understandable but requires minor English polishing. Some sentences are overly long or repetitive. Standardize terminology (e.g., 'non-fermenters' instead of 'no fermenters') and check consistency of abbreviations and reference formatting.

Response 13: We have:

  • Edited long sentences for clarity
  • Standardized terminology to "non-fermenters" throughout
  • Checked all abbreviations for consistency
  • Verified reference formatting according to journal guidelines

Comment 14: The abstract includes duplicated sentences; please remove repetitions.

Response 14: We have removed the duplicated sentence about NDM-producing isolates from the abstract.

Comment 15: Specify whether 'Enterobacterales' and 'non-fermenters' were equally represented in sampling.

Response 15: 'Enterobacterales' and 'non-fermenters' were not  equally represented Page 4 , Paragraph 2

3.1. Phenotypic Identification, Antimicrobial Susceptibility Test (AST) Results, and Carbapenemase Detection 

Ninety-one MDR CPE and Gram-negative non-fermenter isolates were identified phenotypically, including 66 Enterobacterales: Klebsiella pneumoniae strains (n=61), Klebsiella oxytoca strain (n=1), Proteus vulgaris strains (n=2), Escherichia coli strain (n=1), and Enterobacter cloacae strain (n=1), along with 23 non-fermenters (Pseudomonas aeruginosa strains (n=5), Stenotrophomonas maltophilia strains (n=2), and Acinetobacter baumannii strains (n=16),

Reviewer 2 Report

Comments and Suggestions for Authors

The authors study clinical Gram-negative isolates towards cefiderocol. The manuscript can be accepted for publication in Antibiotics after minor revision. The authors should revise the manuscript according to the following comments:

  1. The bacterial strain names should be written in italics. Please check the entire text.
  2. In the Experimental section, the authors mention “25 μl of the bacterial suspension”; however, they should specify the density (CFU/ml) of the suspension.
  3. The authors should present the concentration of cefiderocol in molarity for the agar disk method. Grams are not a concentration unit (check cefiderocol disk (30 μg concentration) were used,)
  4. It is not clear how the susceptibility was determined for each bacterial strain.
  5. In the tables, the confidence limits for MIC and IZ values are missing.

Author Response

RESPONSE TO REVIEWER 2 -Please see the attachment

We thank the reviewer for their thoughtful comments and helpful suggestions. We have carefully addressed each point and revised the manuscript accordingly. Below are our point-by-point responses:

        Comment 1: The bacterial strain names should be written in italics. Please check the entire text.

         Response 1: All bacterial species are now italicized consistently.

Comment 2: In the Experimental section, the authors mention “25 μl of the bacterial suspension”; however, they should specify the density (CFU/ml) of the suspension.

Response 2: Revised text: This protocol requires a standardized bacterial inoculum at a concentration of 0.5 McFarland (equivalent to 1 × 10⁸ CFU/mL), prepared in saline and used for all tests.  Page 12, Paragraph 1.

Comment 3: The authors should present the concentration of cefiderocol in molarity for the agar disk method. Grams are not a concentration unit (check cefiderocol disk (30 μg concentration) were used,)

Response 3: Revised text:

For the Kirby-Bauer disk diffusion (DD) method, Mueller-Hinton agar and a cefiderocol disk (containing 30 µg of active compound, which equals 39.9 nmol per disk). Page 12, Paragraph 4.

         Comment 4: It is not clear how the susceptibility was determined for each bacterial strain.

        Response 4: Added clarification:

        Cefiderocol susceptibility tests

        For each MDR isolate, susceptibility testing for Cefiderocol was performed using EUCAST MIC         breakpoints (BMD) and CLSI zone-diameter criteria (DD) [25]. Page 11, Paragraph 2.

Comment 5:In the tables, the confidence limits for MIC and IZ values are missing.

Response 5: Included 95% CI in Table 4 (Page 6).

Reviewer 3 Report

Comments and Suggestions for Authors

This study addresses the extremely current and important issue of increasing resistance to carbapenems in Gram-negative bacteria and assesses the efficacy of cefiderocol, a modern siderophore cephalosporin. This retrospective analysis provides valuable information on the in vitro activity of cefiderocol against a collection of multidrug-resistant (MDR) Enterobacterales bacteria and non-fermenting bacteria. The study's greatest strength is its high clinical value and relevance to current therapeutic challenges stemming from the growing number of strains resistant to β-lactam antibiotics. The inclusion of various species in the analysis, including Klebsiella pneumoniae, Pseudomonas aeruginosa, Stenotrophomonas maltophilia, and Acinetobacter baumannii, increases the diversity of the research material and strengthens the translational value of the results.

However, the paper requires several additions:
1. It would be worthwhile to include relevant clinical data, such as the age and gender of the patients.
2. Information on the patients' underlying diseases, which are important for the development of infection, such as diabetes and hypertension, is missing.
3. Confirmation of the results using molecular methods would be a valuable addition.
4. A statistical analysis comparing the differences in the percentage of susceptible strains between different bacterial groups (e.g., Enterobacterales vs. A. baumannii) is missing. This would increase the reliability and precision of the conclusions.

Comments on the Quality of English Language

I suggest correction by native

Author Response

RESPONSE TO REVIEWER 3

We thank the reviewer for their thoughtful comments and helpful suggestions. We have carefully addressed each point and revised the manuscript accordingly. Below are our point-by-point responses:

Comment 1: It would be worthwhile to include relevant clinical data, such as the age and gender of the patients.

Response 1:

Our study population comprised elderly individuals, with a median age of 70 and a range of 21 to 93 years. Half of them were women (45). Page 4, Paragraph 1.

Comment 2: Information on the patients' underlying diseases, which are important for the development of infection, such as diabetes and hypertension, is missing.

Response 2: Included

The main comorbidities were hypertension (40.4%), cardiovascular disease (32.6%), and chronic kidney disease (22.5%), along with diabetes mellitus (12.4%); the average was 2.7 comorbidities per patient. Most of the patients were in the Intensive Care Unit (ICU) (n=49, 55%). Page 4, Paragraph 1.

Comment 3: Confirmation of the results using molecular methods would be a valuable addition.

Response 3: Genomic confirmation was not available; limitation acknowledged explicitly, Page 10 Paragraph 2.

This study has several limitations. First, the relatively small sample size (n=89) and single-center design limit how broadly we can apply the findings, but our results align with those from the SENTRY multinational surveillance program regarding reported susceptibility rates among P. aeruginosa isolates resistant to ceftazidime–avibactam [22].. Second, carbapenemase identification relied on immunochromatographic methods without validation by genomic sequencing, potentially missing other resistance mechanisms, such as GES-type carbapenemases or chromosomal mutations.

Comment 4: A statistical analysis comparing the differences in the percentage of susceptible strains between different bacterial groups (e.g., Enterobacterales vs. A. baumannii) is missing. This would increase the reliability and precision of the conclusions

Response 4: Added statistical comparison using the McNemar test with p-values and 95% CI, Page 6, Table 4.

Round 2

Reviewer 1 Report

Comments and Suggestions for Authors

The paper has been improved according to my comments. I have no further comments.

Comments on the Quality of English Language

Overall, the manuscript is understandable but requires minor English polishing. Some sentences are overly long or repetitive. 

Reviewer 3 Report

Comments and Suggestions for Authors

The authors significantly corrected the manuscript according to the reviewer's suggestions. Recently, I recommend the article for publication.

Comments on the Quality of English Language

I suggest correction by native.